# PYL1- and PYL8-like ABA Receptors of *Nicotiana benthamiana* Play a Key Role in ABA Response in Seed and Vegetative Tissue

**DOI:** 10.3390/cells11050795

**Published:** 2022-02-24

**Authors:** Gaston A. Pizzio, Cristian Mayordomo, Jorge Lozano-Juste, Victor Garcia-Carpintero, Marta Vazquez-Vilar, Sergio G. Nebauer, Kacper P. Kaminski, Nikolai V. Ivanov, Juan C. Estevez, Maria Rivera-Moreno, Armando Albert, Diego Orzaez, Pedro L. Rodriguez

**Affiliations:** 1Instituto de Biología Molecular y Celular de Plantas, Consejo Superior de Investigaciones Científicas-Universidad Politécnica de Valencia, ES-46022 Valencia, Spain; gapizzio@gmail.com (G.A.P.); crimaru1@ibmcp.upv.es (C.M.); lojujo@ibmcp.upv.es (J.L.-J.); vicgarb4@upvnet.upv.es (V.G.-C.); marvazvi@ibmcp.upv.es (M.V.-V.); dorzaez@ibmcp.upv.es (D.O.); 2Plant Production Department, Universitat Politècnica de València, ES-46022 Valencia, Spain; sergonne@ebvg.upv.es; 3PMI R&D, Philip Morris Products S.A., Quai Jean Renaud 5, CH-2000 Neuchâtel, Switzerland; kacper.kaminski@pmi.com (K.P.K.); nikolai.ivanov@pmi.com (N.V.I.); 4Centro Singular de Investigación en Química e Bioloxía Molecular (CiQUS), Departamento de Química Orgánica, Universidade de Santiago de Compostela, 15782 Santiago de Compostela, Spain; juancarlos.estevez@usc.es; 5Instituto de Química-Física Rocasolano, Departamento de Cristalografía y Biología Estructural, CSIC, ES-28006 Madrid, Spain; mrmoreno@iqfr.csic.es (M.R.-M.); xalbert@iqfr.csic.es (A.A.)

**Keywords:** ABA receptors, gene editing, CRISPR/Cas9, *Nicotiana benthamiana*, biotechnological crop, drought, multiplex mutations, ABA sensitivity, tetraploid, extremophile

## Abstract

To face the challenges of climate change and sustainable food production, it is essential to develop crop genome editing techniques to pinpoint key genes involved in abiotic stress signaling. The identification of those prevailing abscisic acid (ABA) receptors that mediate plant-environment interactions is quite challenging in polyploid plants because of the high number of genes in the PYR/PYL/RCAR ABA receptor family. *Nicotiana benthamiana* is a biotechnological crop amenable to genome editing, and given the importance of ABA signaling in coping with drought stress, we initiated the analysis of its 23-member family of ABA receptors through multiplex CRISPR/Cas9-mediated editing. We generated several high-order mutants impaired in *NbPYL1-like* and *NbPYL8-like* receptors, which showed certain insensitivity to ABA for inhibition of seedling establishment, growth, and development of shoot and lateral roots as well as reduced sensitivity to the PYL1-agonist cyanabactin (CB). However, in these high-order mutants, regulation of transpiration was not affected and was responsive to ABA treatment. This reveals a robust and redundant control of transpiration in this allotetraploid plant that probably reflects its origin from the extreme habitat of central Australia.

## 1. Introduction

*Nicotiana benthamiana* (*Nb*) is a tobacco species widely used both in research and as a biotechnological crop for high-scale production of different bioproducts. *Nb* originated in the extreme habitat of central Australia, and, accordingly, the laboratory strain is an extremophile plant adapted to harsh growth conditions [1,2]. The *Nicotiana* genus is a part of *Solanaceae*, a family that includes economically relevant plants such as tomato (*Solanum lycopersicum*), potato (*Solanum tuberosum*), eggplant (*Solanum melongena*), and common tobacco (*Nicotiana tabacum*) [3]. The use of Agrobacterium-based transfection in *Nb* has been widely adopted for producing recombinant proteins (e.g., vaccines and antigens) as well as different biopharmaceuticals [4,5,6]. Recently, very efficient reprogramming of *Nb* gene expression has been achieved through vacuum-assisted agroinfiltration as well as delivery of packaged RNA viral vectors by spraying [6]. Therefore, *Nb* has been established as an efficient biotechnological crop and a suitable model plant for studies on agronomic performance [6]. Given the increasing biotechnological relevance of *Nb*, several initiatives have been undertaken to develop genome sequencing and transcriptome analysis of *Nb* [7,8]. *Nb* is an allotetraploid tobacco whose diploid parental progenitors (hybridization event 4–5 million years ago [Mya]) are affiliated with the *Nicotiana* sections *Noctiflorae* and *Sylvestres* [3]. Genomic reads of *N. glauca* (*Noctiflorae*) and *N. sylvestris* (*Sylvestres*) were used to analyze the evolution of the subgenomes in the allotetraploid *Nb*. As a result, Schiavinato et al. [3] found that both subgenomes were in an advanced stage of long-term genome diploidization, lacking conservation of subgenome structure. This contrasts with the young (0–0.5 Mya) allopolyploid *N. tabacum*, in which the two subgenomes of *N. sylvestris* and *N. tomentosiformis* could be clearly separated [9].

The complexity of the *Nb* genome (2n = 38; over 3 Gbp) is high in comparison to the *Arabidopsis thaliana* (At) genome (2n = 10; 0.15 Gbp). This higher complexity is also reflected in the higher number of genes in the former [8]. For example, only 14 members of the PYRABACTIN RESISTANCE1 (PYR1), PYR1-LIKE (PYL), REGULATORY COMPONENTS OF ABA RECEPTORS (RCAR) ABA receptor family (also referred to as PYLs) have been identified in At [10,11,12,13], whereas 23 such functional members were identified in *Nb* (Figure 1; see below). In cultivated tobacco, Bai et al. identified 29 gene sequences that show homology to ABA receptor genes [14]. Given the recent hybridization event of the allotetraploid tobacco *N. tabacum*, the authors could identify 11 and 16 PYLs in the genomes of its two diploid ancestors, *N. tomentosiformis* and *N. sylvestris*, respectively [14]. Few studies have investigated the physiology of *Nb*, and efforts for annotation and analysis of its gene families are just starting [8]. In this work, we have investigated the large family of PYR/PYL/RCAR ABA receptors in *Nb* and generated and physiologically characterized CRISPR/Cas9-edited *Nb* mutants that are impaired in several of the major ABA receptors.

ABA signaling plays a critical role in plant response to abiotic stress. Given that climate change is proving a challenge for plant agriculture, genetic studies in crops are needed to unveil further the role of ABA in agronomic adaptation to environmental stress and to pinpoint the most relevant receptors for drought response. Thus, our study aims to cover two major goals: starting to dissect a complex gene family of hormone receptors and generating mutants that are impaired in ABA signaling in a crop. Only one study to date has reported the generation of high-order *pyl* mutants in a crop (i.e., rice) [15]. In this case, the study was mostly focused on mutations in a subfamily of ABA receptor genes to enhance rice growth and productivity in paddy field conditions, in which water is not limited [15]. Interestingly, some combinations of high-order *pyl* mutants (e.g., *pyl1/4/6*) in this study showed improved growth compared to wild type (WT), which indicates that genetic adjustment of ABA signaling might help generate improved crop varieties in certain environments [15]. Similar studies have not yet been performed on other crops. Therefore, given the capital importance of ABA signaling for plant physiology and the wide expansion of genome editing technologies, we decided to investigate the PYR/PYL/RCAR family of ABA receptors in *Nb* by generating CRISPR/Cas9-edited mutants. Future studies could take advantage of this knowledge by using the prevailing ABA receptors to engineer drought-tolerant crops.

## 2. Materials and Methods

### 2.1. Plant Material and Growth Conditions

The *N. benthamiana* LAB (laboratory) strain was stably transformed with *Agrobacterium tumefaciens* LBA4404 cells harboring the indicated binary vector. Biallelic (BA) and homozygous mutants were used for phenotype analysis. The edited genes and mutations in the T1 lines are described in Table 1. The high-order mutants used in this study are described in Table 2. The plants were grown under a 16-h light, 24 °C constant temperature/8-h dark regimen at a constant 20 °C temperature.

### 2.2. Sample Preparation and RNA-seq Analysis

In plants grown for 4 weeks, the whole 5th leaf from the main axis was collected from several plants. Three biological replicates were made, each coming from a pool of 3 leaves. The biological replicates were ground in liquid nitrogen and stored at −80 °C. RNA was extracted with the GeneJET RNA purification kit (Thermofisher). The extracted RNA samples were prepared with the Universal Plus mRNA-Seq Library Preparation kit through NuQuant. The libraries were later sequenced with an Illumina NovaSeq^®^ 6000 system. Paired-end (PE) 2 x 150 bp sequencing was performed with the NovaSeq6000-Dual Index-Paired End-S4-XP protocol. The sequencing data thus generated were demultiplexed by Illumina BaseSpace^®^ Clarity LIMS (Illumina, Inc., San Diego, CA, USA).

The sequence reads were quality checked using FastQC v0.11.9. Raw reads were quality trimmed, and Illumina adaptors were removed with Trimmomatic v0.39. The reads were mapped against *N. benthamiana* genome version v3.3 (provided by the *Nicotiana benthamiana* consortium; https://www.nbenth.com (accessed on 30 January 2021)) by using HISAT2, and gene abundances were calculated by using StringTie and version 3.02 of gene annotation for this assembly. From these counts, a gene expression table was generated for the *PYL*/*PYR* genes in FPKM (fragments per kilobase of exon model per million mapped reads) units. The RNA-seq data reported in this article are available in the NCBI database (Bioproject PRJNA770890).

### 2.3. CRISPR/Cas9 Vector Construction

Cloning was performed by using the multiplex RNA-guided CRISPR/Cas9 system adapted for GoldenBraid [16]. We generated polycistronic gRNAs, structured into 3 TUs, which overall contained 10 gRNAs that potentially targeted 7 ABA receptors: *g66232*, *g59819*, *g78400*, *g90182*, *g91686*, *g2543*, and *g110177*. For each gene, 2 target sequences were selected (20 nucleotides each gRNA; PAM sequence NGG; Table 2 and Appendix A), which were obtained with CRISPRdirect (http://crispr.dbcls.jp (accessed on 30 November 2019)) by using the *N. benthamiana* genome sequence v1.0.1 (July 2014) as reference. We selected target sequences with no off-targets in the 20-mer sequences and 5 or fewer off-targets in the 12-mer sequences, including an optimum GC percentage (40–65%). Given the high degree of identity between *PYL1a* (g90182) and *PYL1b* (g91686), gRNAs number 1 and 6 served to target both genes. Likewise, for *PYL8b* (*g59819*) and *PYL8a* (*g78400*), gRNAs number 2 and 7 were selected for inactivating both genes.

After selecting the target sequences, we created each GB-adapted gRNA part by using the CRISPR domesticator tool (https://gbcloning.upv.es/do/crispr/cas9_multiplexing (accessed on 30 November 2019)). We produced a set of 10 domesticated gRNA sequences flanked by standard GB overhangs (prefix and suffix; g1 to g10) by using two semi-complementary oligonucleotides for each gRNA (Table 2). The oligonucleotides were diluted to a final concentration of 1 µM, mixed, and annealed for 30 min at room temperature and then used for the multipartite reaction. For assembly of gRNAs on level 0, 2 complementary primers (2 ng each) for each gRNA (g1–g10) were included in a *BsmB*I restriction–ligation reaction together with 75 ng of pUPD2 and 75 ng of the corresponding level 1 tRNA-scaffold plasmid depending on the desired position of each target on the level 1 assembly. All level 1 gRNA constructs were confirmed by restriction analysis and subsequent sequencing.

Polycistronic gRNA assemblies (multiplex) on level 1 were performed through multipartite reactions to obtain TUs from basic domesticated level 0 parts. The correspondent pUPD2–sgRNA constructs (g1–g10) were included in a *Bsa*I restriction–ligation reaction together with 75 ng of GB1001 (Pol III promoter U6-26) and 75 ng of pDGB3_alpha1 or pDGB3_alpha2. Thus, 3 TUs were obtained: pU6-26_g8g5g1_DGB3_α1, pU6-26_ g10g6g2_DGB3_α1, and pU6-26_ g9g7g4g3_DGB3_α2. All level 1 gRNA constructs were validated by restriction analysis and confirmed by sequencing. Several TUs were combined on level > 1 with bipartite *BsmB*I- or *Bsa*I-mediated reactions to create modules (Goldenbraid 2.0 system) [17]. First, TUs pU6-26_ g10g6g2_DGB3_α1 and pU6-26_ g9g7g4g3_DGB3_α2 were combined into a pDGB3_Ω2 vector. Then, GB2235 (pNOS::NPTII/35S::hCAS9/35S::DsRED) was combined to generate a pDGB3_α2 vector. Finally, the pU6-26_ g8g5g1_DGB3_α1 vector was combined to generate a final pDGB3_Ω2 vector. The constructs were confirmed by restriction analysis and subsequent sequencing.

### 2.4. In Vitro Culture for Stable Transformation of N. benthamiana

Transformation and in vitro regeneration were achieved by the leaf-disc method [18], which was followed with minor modifications as described by Vazquez-Vilar et al. [19]. Briefly, fully expanded leaves of 6-week-old *N. benthamiana* plants were sterilized for 10 min in a 5% commercial bleach solution supplemented with 20 µL/L Tween-20. The sterile leaves were immersed in 70% ethanol and immediately transferred to sterile demi water for 4 consecutive washing steps. The sterile leaves were then placed on sterile filter paper, and leaf discs were excised with a 0.5-cm-diameter sterile cork-borer. The leaf discs were incubated for 15 min with an *A. tumefaciens* LBA4404 strain carrying the final vector described in the previous section. The *A. tumefaciens* cells were grown overnight to an OD600 of 0.5 in TY medium (5 g/L triptone and 3 g/L yeast extract) supplemented with 1 mM MgSO_4_·7H_2_O, 1 mM MgCl_2_, and 200 µM acetosyringone. After incubation with *A. tumefaciens*, the leaf discs were placed in cocultivation medium plates (Murashige Skoog [MS] medium supplemented with vitamins [Duchefa], 3% sucrose, 0.8% phytoagar, 1 mg/L 6-benzylaminopurine, and 0.1 mg/L 1-naphtalenacetic acid) and incubated for 2 days in the dark. Next, the discs were transferred into selection medium (cocultivation medium with 200 mg/L carbenicillin and 100 mg/L kanamycin) and then changed to fresh medium approximately every 10 days until calli (first) and shoots (next) appeared (6–8 weeks). Individual shoots were excised and transferred into rooting medium (MS medium supplemented with vitamins [Duchefa], 3% sucrose, 0.8% phytoagar, and 100 mg/L kanamycin) until roots appeared. In vitro culture was performed under long-day conditions in a growth chamber (16-h light/8-h dark; 24 °C; 60–70% humidity; 250 µmol/m^2^ s photons).

### 2.5. Analysis of Edited Genomic DNA and Identification of Mutant Alleles

To analyze the mutations caused by our CRISPR/Cas 9-mediated editing efforts, genomic DNA was extracted from the leaves (150 mg of tissue) of each transgenic plant by using the cetyltrimethylammonium bromide (CTAB) method [20]. Amplicons containing the CRISPR target site were amplified by PCR by using specific primers (Appendix A), cleaned with ExoSAP-IT (ThermoFisher), and sequenced. Chromatograms of edited genomic DNA were analyzed using Inference of CRISPR Edits (ICE) from Synthego.

### 2.6. ABA Sensitivity in Seedling Establishment Assays

After surface sterilization of seeds, stratification was performed in the dark at 4 °C for 3 days. Seeds of the wild-type *N. benthamiana* LAB strain and the indicated mutants were sown on MS plates supplemented with different concentrations of ABA or CB per experiment. Seedling establishment was scored by the percentage of seeds that developed green expanded cotyledons and the first pair of true leaves. Shoot projected area was measured as an indicator of seedling growth.

### 2.7. Root and Shoot Growth Assays

After seed germination, the seedlings were grown on vertically oriented MS plates for 4 days. They were then transferred to new MS plates lacking or supplemented with the indicated concentrations of ABA. The plates were scanned on a flatbed scanner after 14 days to obtain image files suitable for quantitative analysis of root growth by using the NIH software ImageJ v1.37. As parameters for measuring overall root growth, PR length and the total number and length of secondary roots were quantified. As an indicator of shoot growth, FW and projected area of the leaves were measured.

### 2.8. Expression of NbPYL1b in E. coli, Purification of His-Tagged Protein, and Gel Filtration Analysis

The open reading frame of *NbPYL1b* was amplified by PCR by using *Nb* genomic DNA as a template and then cloned into pCR8/GW/TOPO. The ORF was then excised by *Nco*I–*Hind*III double digestion and subcloned into pETM11. Escherichia coli BL21 (DE3) cells were transformed with the corresponding pETM11-*NbPYL1b* construct and grown at 30 °C to an optical density at 600 nm of 0.6 in 100 mL of Luria–Bertani medium supplemented with 50 μg mL^−1^ kanamycin. Then, 1 mM isopropyl-β-d-thiogalactopyranoside (IPTG) was added to the medium, and the cells were further grown for 16 h at 16 °C. After induction, the cells were harvested by centrifugation and stored at −80 °C until protein purification. The cell pellet was resuspended in 2 mL of HIS buffer (50 mM Tris HCl [pH 7.6], 250 mM KCl, 10% glycerol, 0.1% Tween-20, and 10 mM mercaptoethanol), and the cells were sonicated in a Branson sonifier. A clear lysate was obtained after centrifugation at 14,000× *g* for 15 min, and it was diluted with 2 volumes of HIS buffer. This protein extract was applied to a 0.5-mL nickel–nitrilotriacetic acid (Ni-NTA) agarose column and the column was washed with 10 mL of HIS buffer supplemented with 20% glycerol and 30 mM imidazole. Bound protein was eluted with HIS buffer supplemented with 20% glycerol and 250 mM imidazole. Protein concentration was calculated using the Bradford assay (BIORAD), and protein quality was checked by 12% SDS-PAGE.

For gel filtration analysis, the protein was purified from a 400 mL culture, and the His tag was cleaved using tobacco etch virus protease. The protein was finally eluted using 30 mM Tris pH 7.5, 150 mM NaCl, 60 mM imidazole, and 1 mM DTT buffer and concentrated to 2.7 mg/mL. This sample was injected on a HiLoad Superdex 200 16/60 column (GE Healthcare) previously equilibrated with 30 mM Tris pH 7.5, 150 mM NaCl, 1 mM DTT buffer. A calibration curve was obtained for the column by plotting the molecular weight of 5 protein standards against their Kav (proportion of pores available to the molecule).

### 2.9. PP2C Activity Assay

Phosphatase activity was measured using p-nitrophenyl phosphate (pNPP) as a substrate. The assays were performed in a 100 μL solution containing 25 mM Tris HCl [pH 7.5], 2 mM MnCl_2_, and 25 mM pNPP. The assays contained 0.5 μM phosphatase (ΔN-HAB1), 1 or 2 μM receptor (NbPYL1b), and the indicated concentrations of ABA or CB. Phosphatase activity was recorded with a ViktorX5 reader at 405 nm every 60 s over 30 min, and the activity obtained after 30 min as indicated in the graphs. CB was synthesized from commercial 4-amino-2-chlorobenzonitrile through a Suzuky coupling with commercial cyclopropylboronic acid followed by a sulfonamide coupling with commercial 4-methylbenzenemethanesulfonyl chloride, using the protocols specified in literature [21].

### 2.10. Measurement of Stomatal Conductance (Gs) and Transpiration Rate (E)

The LICOR-6400 system (LICOR Biosciences, Lincoln, NE, USA) was used to measure stomatal conductance and transpiration rate in 28-day-old plants grown in the greenhouse [22]. At 2 h after the beginning of the light period, instantaneous measurements (n = 5) were taken under steady-state conditions at saturating light (1000 μmol m^−2^ s^−1^), 400 ppm CO_2_, 1–2.5 kPa vapor pressure difference, and ambient temperature (25–27 °C). The effect of ABA on regulation of stomatal closure in the mutant lines was estimated by spraying 50 μM ABA solution on adaxial and abaxial sides of leaves. This application was performed 18 h before gas exchange measurements.

### 2.11. Statistical Analyses

Statistical analysis was performed using GraphPad Prism. One-way ANOVA (followed by Dunnet’s test) or Student’s *t*-test was performed to determine significant differences between groups of samples. Levels of significance are indicated in the figures by asterisks: * *p* ≤ 0.05; ** *p* ≤ 0.01; *** *p* ≤ 0.001. Values are plotted as mean ± SE and were obtained from at least 2 independent experiments.

## 3. Results

### 3.1. Genome-Wide Analysis of ABA Receptors in Nb

To identify ABA receptors in the genome of Nb, we performed a BLAST search using as queries the amino acid sequences of the 14 At PYR/PYL/RCAR ABA receptors. We identified 28 gene sequences whose gene products showed similarity to ABA receptors. However, five gene products likely lead to non-functional receptors because of severe truncations or the presence of nonsense mutations (Figure 1, indicated in grey).

Therefore, we concentrated on the remaining 23 gene sequences and performed a phylogenetic analysis, including protein sequences for both AtPYLs and NbPYLs (Figure 2). As expected from previous analyses performed in At and different crops [10,11,23,24], we identified three major subfamilies of *Nb* ABA receptors, which we named as PYL1/2-like (subfamily III), PYL4/5-like (subfamily II), and PYL8/9-like (subfamily I). ABA receptors are very similar to the protein level [10,11,12,13]; however, subfamily III is composed of dimeric receptors with lower intrinsic affinity for ABA, whereas subfamilies I and II comprise monomeric receptors that show a higher intrinsic affinity for ABA. Dimeric receptors require dimer dissociation in response to ABA during the receptor activation process, whereas monomeric receptors lack this thermodynamic penalty for interaction with clade A PP2Cs [12]. In *Nb*, our analysis did not detect any member of the AtPYL11-13 subfamily. Our findings showed a good correlation with the current data on ABA receptors in *N. tabacum* reported by Bai et al. [14].

### 3.2. RNA-seq Analysis and Target Selection for Genome Editing Using CRISPR/Cas9 Technology

RNA-seq data for *Nb* gene expression in different plant tissues are just emerging [8]. We took advantage of the data obtained in the whole fifth leaf from the main axis to identify the most expressed receptors in this tissue (Figure 3A).

Thus, two genes of subfamily III (*PYL1a* and *PYL1b*) and one gene of subfamily I (*PYL8a*) showed the highest expression among all *NbPYL* members. In this tissue, members of subfamily II showed lower expression than other subfamilies. Genome sequencing updates, annotation studies, and RNA-seq analyses are still underway in this field. Therefore, in the future, further inspection should be performed to update the expression profile of *NbPYLs* in different tissues. We constructed CRISPR/Cas9 vectors to edit the most expressed receptors according to our RNA-seq data, i.e., *PYL1a*, *PYL1b,* and *PYL8a*, as well as other representative members of subfamilies I and II (Figure 3B; Appendix A). Some ABA receptors escaped detection in our initial analysis because the genome annotation was updated and new data emerged while this work was under process (see current gene coordinates and protein sequences in Appendix A, https://www.nbenth.com/ (accessed on 30 November 2021)); for example, *PYL8d* is a highly expressed receptor that was not tagged in our CRISPR/Cas9 constructs (see below).

To edit the receptors described above as well as other genes including *PYL4b*, *PYL4d*, *PYL8b,* and *PYL8c*, we generated polycistronic guide RNAs (gRNAs) constructs, tandemly arrayed, to target them. Thus, we generated a 10-gRNA construct structured into three transcriptional units and potentially targeting seven ABA receptors (Figure 3B). However, in our hands, the editing efficiency decreased dramatically from the last position of the polycistronic gRNAs, which turned out to be the only effective position for generating high-order mutants in the germinal line (Figure 3B). Thus, we were able to generate mutant alleles in the germinal line in only five ABA receptors belonging to subfamilies I and III (Table 1), whereas we could not obtain mutant alleles of subfamily II receptors. Upon selecting homozygous individuals, we obtained four high-order mutants, including two quadruple and two triple mutants (Table 2). Sequence analysis and comparison of the chromatograms of WT and mutant plants showed different insertions (+1) or deletions (∆1, ∆2, ∆4, and ∆24) in the mutant alleles (Table 1; Appendix A).

### 3.3. Biochemical Analysis of NbPYL1b

The structure of the PYR/PYL/RCAR ABA receptors shows the classical fold of the START/Bet v proteins [12,25,26,27,28]. This consists of an α/β helix-grip fold with seven antiparallel β-sheets that form a U-shaped hydrophobic cleft flanked by amino- and carboxyl-terminal α-helices [29]. Thus, a large central hydrophobic pocket is generated that serves to accommodate ABA. PYR/PYL/RCARs alone are able to bind ABA, but only in the presence of the protein phosphatase type-2C (PP2C) co-receptor are able to bind the ligand with nanomolar affinity [10,12]. We performed an amino acid sequence alignment using the software ESPRIPT to correlate the secondary structure of the Nb receptors with the crystallographic data of At receptors, specifically with the available crystal structure of AtPYL1 (Figure 4A). We thus confirmed that NbPYL1a, PYL1b, PYL8a, PYL8b, and PYL8c contain a bona fide structural arrangement as previously described in ABA receptors. The conserved residues of the loops found between the β3-β4 and β5-β6 regions, corresponding to the gate and latch loops, respectively [29], are fully conserved in the above NbPYLs.

Inhibition of clade A PP2Cs in a ligand-dependent manner is the key biochemical feature of ABA receptors. Therefore, we decided to obtain recombinant protein for NbPYL1b (whose amino acid sequence is 98% identical to that of NbPYL1a) and test its activity. We produced a recombinant His-tagged protein, purified it by Ni-NTA affinity chromatography, and found that NbPYL1b showed ABA-dependent inhibition of HAB1 (Figure 4B, left panel). Even at a 1:4 ratio (phosphatase:receptor), HAB1 inhibition was dependent on the presence of the ABA ligand. Additionally, we tested another ligand, the ABA receptor-agonist cyanabactin (CB), which has been reported to inhibit Arabidopsis PYL1 [21]. Interestingly, CB was also a good ligand for NbPYL1b, and its half-maximal inhibitory concentration (IC50) against HAB1 activity was even lower than that of ABA (Figure 4B, right panel). This result also confirms that synthetic ligands of At ABA receptors are active against crop receptors (Figure 4B). Finally, we analyzed the oligomeric state of NbPYL1b to test whether its amino acid sequence similarity to AtPYL1/AtPYR1 also leads to a dimeric structure. Both AtPYR1 and AtPYL1 are homodimeric in the absence of ABA, and ligand binding induces dissociation of the dimer to form a 1:1 monomeric complex with the PP2C co-receptor [12,25,26,27,28]. Interestingly, analysis by gel filtration chromatography indicated that NbPYL1b was eluted as a dimer (Figure 4C). Therefore, the inhibition of PP2C activity by NbPYL1b in response to ABA or CB ligands (Figure 4B) reflects the formation of a receptor-ligand-phosphatase complex after dissociation of the dimeric NbPYL1b.

### 3.4. Genetic Analysis of NbPYLs

Even though gene redundancy exists in the PYR/PYL/RCAR family, specific functions and distinctive biochemical features have been reported for several AtPYLs, for example, in growth regulation of the primary and lateral roots, stabilization by ligand binding, stomatal closure, leaf senescence and plant immunity [30,31,32,33,34,35] as well as for the dimeric receptor BdPYL1 in Brachypodium [36]. However, polyploid plants (e.g., benthamiana, wheat, soybean, potato, strawberry, sugarcane, cotton, and canola) might show higher functional redundancy in ABA signaling because of their higher number of PYR/PYL/RCAR receptors [14,37]. Therefore, we decided to analyze several ABA responses in WT and mutant plants to investigate the possible functional redundancy of ABA receptors in Nb.

Initially, we performed seedling establishment assays in the presence of 1 μM ABA and found that the *pyl1b pyl8a pyl8b pyl8c* quadruple mutant was less sensitive to ABA-mediated inhibition of seedling establishment and early seedling growth than the WT (Figure 5A). Seedling establishment of the *pyl1b pyl8a pyl8b* triple mutant was also less sensitive to ABA-mediated inhibition than that of the WT (Figure 5A). Given that CB is a good agonist for PYL1 receptors and specifically for NbPYL1b (Figure 4b), we assayed a quadruple mutant that lacks both *pyl1a* and *pyl1b*, as well as *pyl8a* and *pyl8b*. Thus, we compared the inhibition of seedling establishment by CB in WT and plants in a dose-response assay. We found that seed germination and seedling establishment in the *pyl1a pyl1b pyl8a pyl8b* was less sensitive to inhibition by CB than in the WT (Figure 5B).

At high concentration, ABA inhibits primary root (PR) growth as well as the development and growth of lateral roots (LRs) [15,38]. We scored the ABA-mediated inhibition of PR growth and LRs with 10 μM ABA in *pyl1b pyl8a pyl8b pyl8c* and WT plants (Figure 6).

While the mutant and WT plants showed no significant difference in inhibition of PR growth, the number of LRs, as well as their growth, was higher in the quadruple mutant than in the WT plant, which indicates that the mutant shows a reduced sensitivity to ABA in this response as well as a functional redundancy in ABA signaling for inhibition of PR growth (Figure 6). Similar results were observed both in PR and LR growth in the *pyl1a pyl8a pyl8b* and *pyl1b pyl8a pyl8b* triple mutants in ABA-supplemented medium (Appendix A). In the xerobranching response of arabidopsis and barley, it has been described that LR initiation is temporally inhibited to avoid the development of LRs in the air when roots are not in contact with water and that this response is dependent on ABA signaling [38]. Accordingly, sextuple and septuple pyl mutants of *At* are resistant to ABA-mediated inhibition of LR growth [38]. Our present results in *Nb* confirm that ABA receptors play a key role in mediating the inhibition of LR growth. Interestingly, upon sustained growth in medium supplemented with ABA, the quadruple mutant showed less inhibition of leaf growth than the WT plant (Figure 7). The fresh weight (FW) of leaves was less reduced by ABA in the quadruple mutant than in the WT plants (Figure 7). Likewise, the *pyl1a pyl8a pyl8b* triple mutant was less sensitive to ABA than the WT plant in this assay (Figure 7).

ABA signaling plays a crucial role in regulating stomatal aperture. Increased transpiration and, accordingly, reduced leaf temperature have been reported in mutants impaired in ABA signaling [39,40]. However, when we measured the stomatal conductance and transpiration rate in mock- and ABA-treated adult plants, we did not detect any significant difference between WT and mutant plants (Table 3). ABA treatment induced stomatal closure (measured 18 h after spraying the leaves with a 50 μM ABA solution) and reduced transpiration rates. Accordingly, the plants showed a higher leaf temperature owing to the effect of ABA treatment. Nevertheless, no differences were observed between the WT and mutant genotypes in the ABA-treated plants. This suggests that additional ABA receptors in *Nb* can regulate transpiration in the absence of the PYL1-like and PYL8-like receptors that were inactivated in these mutants. Therefore, redundant control of transpiration by ABA signaling occurs in *Nb*.

## 4. Discussion

Sequencing of a high number of plant genomes has revealed that many angiosperms have undergone polyploidization events along with their evolutionary history [41,42]. Indeed, polyploidy is considered a driving force in plant evolution and adaptation [43], and either auto- or allopolyploidization events have contributed to speciation and diversification [42]. Recent research has revealed the adaptive effect of polyploidy on plant response to global climate change as well as improving tolerance to plant pathogens [43,44]. Polyploidy also leads in woody plants to larger xylem and phloem conduits, lowering hydraulic resistance and facilitating long-distance transport between organs [45]. Particularly, polyploidy can lead to adaptive variation within species in order to help a species colonize harsh geographical niches, and it seems to confer an advantage for confronting abiotic stress conditions [43,46]. The laboratory strain of *Nb* is an extremophile that originated in the extreme habitat of central Australia [1,2]. The harsh conditions of this habitat have likely exerted selection pressure to maintain strict control of the plant water balance, which can be facilitated by a double set of ABA receptors in the genome. Therefore, even though we inactivated several of the most expressed ABA receptors, the *pyl* mutants described here still showed an ABA response for regulation of stomatal aperture (Table 3).

Research on dissecting ABA signaling in crops is just starting, and CRISPR/Cas9-mediated gene editing of ABA receptors has only been described in rice (diploid *Oryza sativa*) thus far [15]. In order to start genetic dissection of the PYR/PYL/RCAR family of ABA receptors in *Nb*, we designed a 10-gRNA construct to potentially target 7 ABA receptors. However, the editing efficiency decreased dramatically from the last position of the polycistronic gRNAs, and we could finally tag only 5 ABA receptors. The CRISPR/Cas9 technology follows an explosive advance in genome editing approaches, and new approaches have been described for improving multiplexed strategies, which are particularly required in polyploid species. For example, Stuttmann et al. [47] have described a highly efficient multiplex editing approach that is based on individual Pol III promoters for gRNA expression. Future approaches for further dissecting the *Nb* family of ABA receptors could take advantage of the genetic resources generated hereby to facilitate additional editing of this 23-member family. For example, additional inactivation of *NbPYL1c*, *NbPYL8d,* and some members of subfamily II in the high-order mutants described here might further unravel functional redundancy for ABA signaling in the regulation of transpiration.

Although we targeted only 5 ABA receptors, we were able to document different phenotypes that revealed reduced ABA sensitivity in the mutants thus generated. This can be explained by the fact that we were able to inactivate some of the major ABA receptors expressed in leaves, and these receptors are also probably actively expressed in other tissues such as LRs or at the seed/seedling stage. Thus, we observed reduced sensitivity to ABA-mediated inhibition of seedling establishment, development of LRs, and inhibition of shoot growth. Conversely, the ABA-mediated inhibition of root growth (at high concentration) and ABA-induced stomatal closure, measured as reduction of stomatal conductance, were not significantly different in the mutants generated in this study relative to WT. We suggest that functional redundancy precludes the appearance of a distinctive phenotype in these assays because only 5 out of 23 receptors were impaired. In order to unveil any functional redundancy, more RNA-seq data are required to pinpoint additional ABA receptors for gene editing. For example, RNA-seq studies focused on root tissue or guard cells should provide clues for targeting additional receptors that are important for regulating root growth and transpiration. Finally, another tool for unveiling any functional redundancy in the complex family of ABA receptors is the use of ABA agonists that show specificity for a particular set of ABA receptors. For example, we took advantage of CB, a rationally designed ABA agonist that is particularly active on PYR1/PYL1-like ABA receptors [21]. Given that one of our *pyl* mutants was impaired both in NbPYL1a and NbPYL1b, two major PYL1-like receptors in *Nb*, we could detect the reduced sensitivity of this mutant to CB (relative to WT) in seedling establishment assays.

## Figures and Tables

**Figure 1 cells-11-00795-f001:**
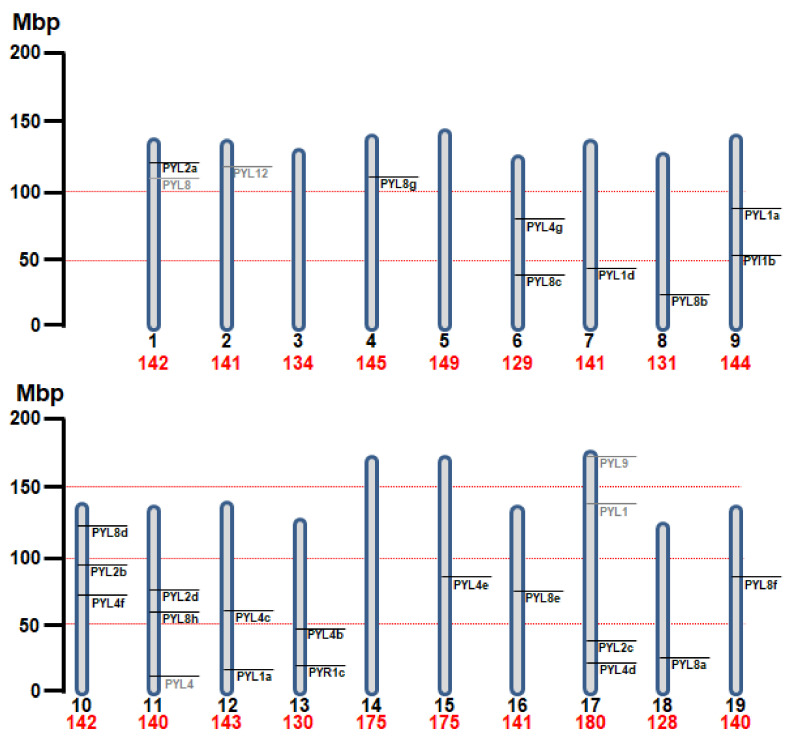
Location of the ABA receptor genes in the chromosomes of *Nicotiana benthamiana* (n = 19). The size (Mbp) of each chromosome is indicated in red and shown by its relative length. The position of each ABA receptor gene is indicated by bars. Those genes labeled in grey indicate sequences that show homology to ABA receptors, but whose gene products likely lead to non-functional receptors because of severe truncations or the presence of nonsense mutations. The figure was generated using the genome coordinates indicated in Appendix A according to www.nbenth.com (v3.3) (accessed on 30 November 2021).

**Figure 2 cells-11-00795-f002:**
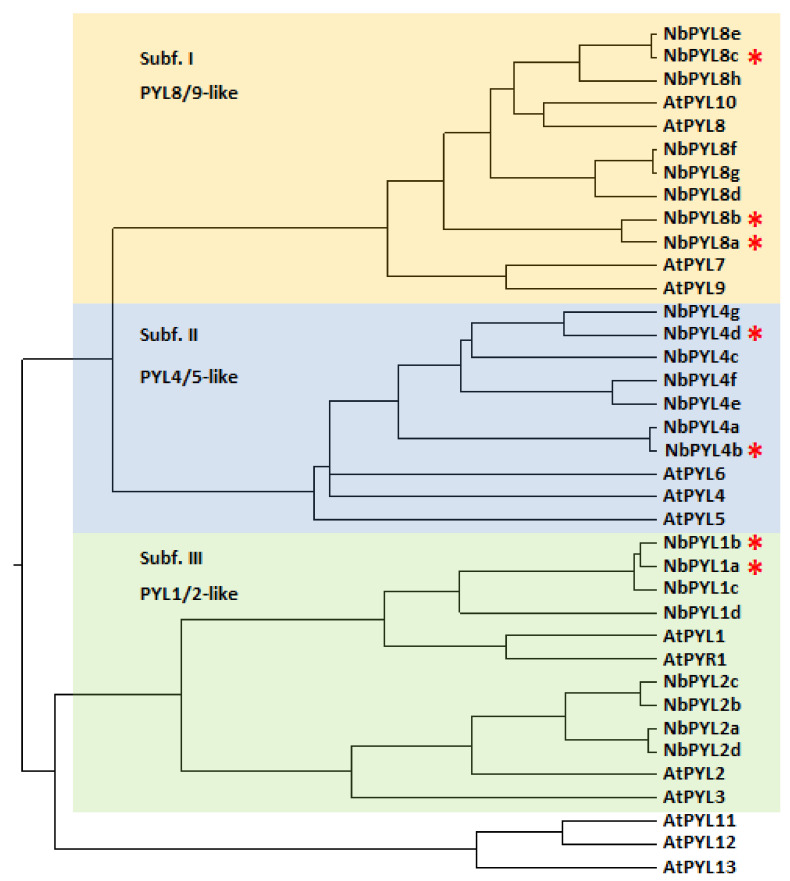
Phylogenetic analysis of PYL proteins in *Arabidopsis thaliana* and *Nicotiana benthamiana*. A total of 14 AtPYLs (PYR1 to PYL13) and 23 NbPYLs were included to generate the cladogram by using the MegAlign 7.0 software. The NbPYL proteins were clustered in three subfamilies as previously described for PYLs in other plant species [24], including subfamily III of putative dimeric receptors. NbPYLs orthologous to AtPYL11-13 were not detected. Red asterisks mark the genes described in this work. An alphanumeric descriptor has been included for each PYL receptor to facilitate naming.

**Figure 3 cells-11-00795-f003:**
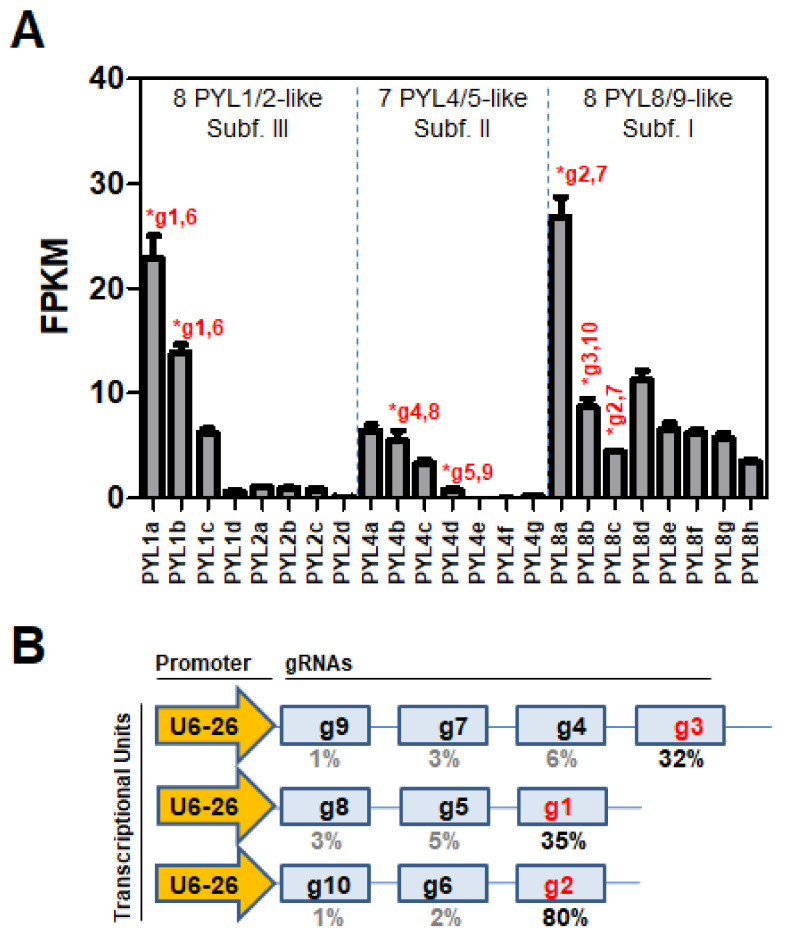
Relative expression of *NbPYLs* in the fifth leaf and scheme of CRISPR/Cas9 guide RNAs (gRNAs) used in this study. (**A**) The gene expression of *NbPYLs* was determined by RNA-seq analysis. Data for mRNA expression of ABA receptors in leaves was normalized and expressed in FPKM (Fragments per kilobase of exon model per million mapped reads). Target genes selected for CRISPR/Cas9-mediated editing are indicated by red asterisks, including the number (see below Figure 3B) of the gRNAs designed for each receptor gene. (**B**) Structure of the polycistronic transcriptional units in the GoldenBraid-based plasmid construct indicating the position of each gRNA after the U6-26 promoter. The last position (indicated in red) of each transcriptional unit turned out to be the most effective for gene editing. The efficiency of generating edited alleles is indicated below as the percentage of mutations found after sequencing of T1 plants. Given the high degree of identity between PYL1a and PYL1b, or PYL8a and PYL8b, the gRNAs 1 and 6, or 2 and 7, served to target both pairs of genes, respectively.

**Figure 4 cells-11-00795-f004:**
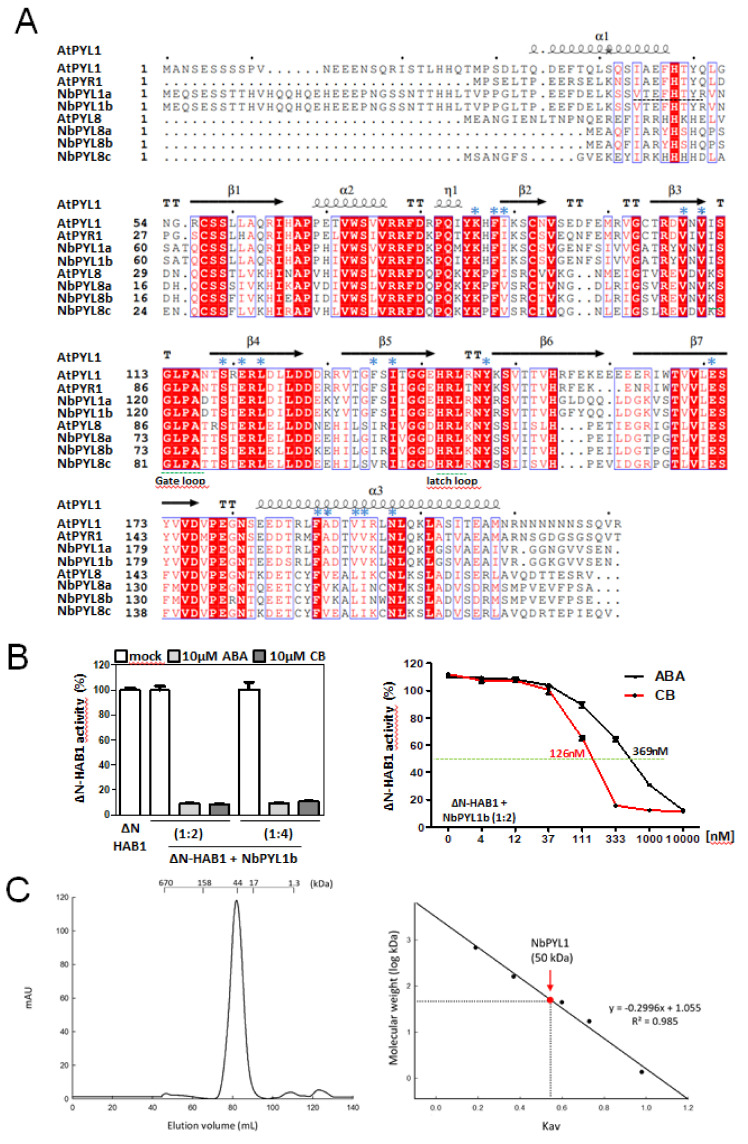
Amino acid sequence alignment of AtPYL1, AtPYR1, AtPYL8, and five NbPYLs. (**A**) Sequence and secondary structure alignment of ABA receptors are indicated. The secondary structure of the NbPYLs is predicted according to the crystallographic structure of AtPYL1 (Protein DataBank Code 3JRS), and was generated using the ESPRIPT program (http://espript.ibcp.fr/ESPript/ESPript/ (accessed on 16 January 2022)). Boxes indicate the position of the gate and latch loops. Blue asterisks mark residues involved in interactions with ABA. The dashed line indicates the 8 amino acid residues that are predicted to be deleted in the *pyl1a-2* allele. (**B**) Purified recombinant His-tagged NbPYL1b shows ABA-dependent inhibition of the PP2C HAB1. Left panel, CB- and ABA-dependent inhibition of the PP2C HAB1 assayed using pNPP as a substrate. Right panel, determination of the IC50 for inhibition of HAB1 by CB or ABA and NbPYL1b. (**C**) The elution profile of NbPYL1b after gel filtration chromatography indicates that the protein was eluted as a dimer. The positions of the molecular weight markers are indicated. The inset represents the calibration curve for the column, obtained by plotting the molecular weight of five protein standards against their Kav (proportion of pores available to the molecule) for this column (black dots).

**Figure 5 cells-11-00795-f005:**
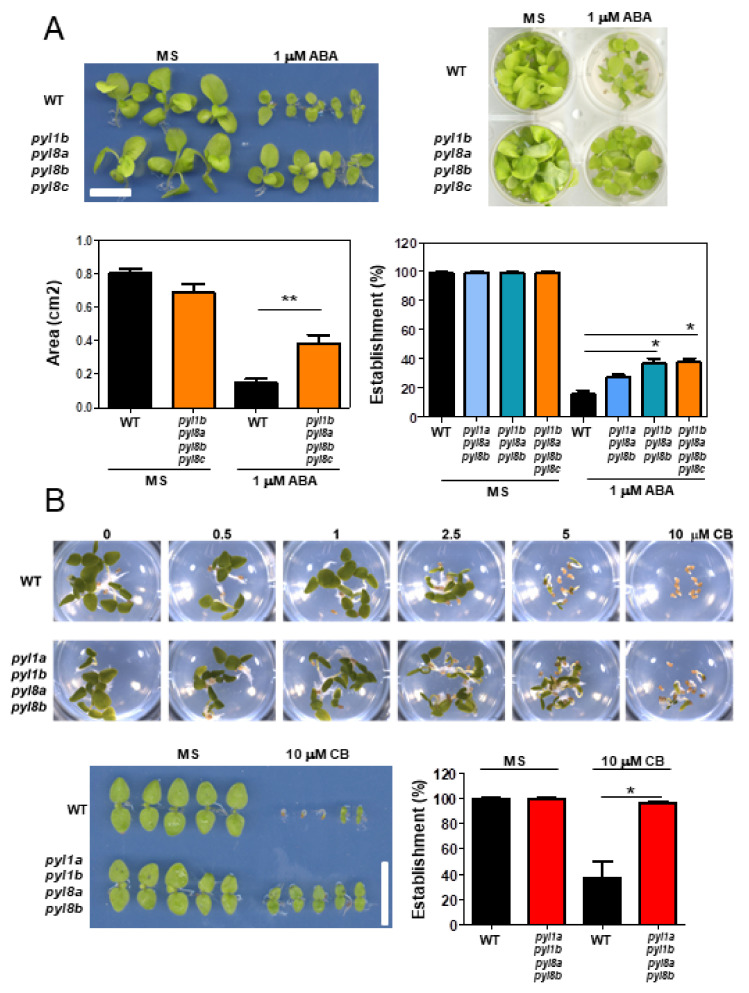
Reduced sensitivity to ABA or the ABA receptor-agonist CB in *Nb* high-order pyl mutants. (**A**) ABA-mediated inhibition of seedling establishment and early seedling growth is less in *pyl1b pyl8a pyl8b pyl8c* than in the WT. Seeds of the WT and mutant plants were germinated in MS medium lacking or supplemented with 1 μM ABA. Seedling establishment was scored after 9 days, and early seedling growth was scored after 15 days. The growth of the seedlings was quantified using IMAGE-J to obtain seedling projected area as a parameter for measuring growth. (**B**) CB-mediated inhibition of germination and seedling establishment is less in *pyl1a pyl1b pyl8a pyl8b* than in the WT. The asterisk indicates: * *p* ≤ 0.05 and ** *p* ≤ 0.01 (Student’s *t*-test or ANOVA followed by Dunnet’s test) when comparing data of mutant lines with non-transformed WT plants in the same assay conditions.

**Figure 6 cells-11-00795-f006:**
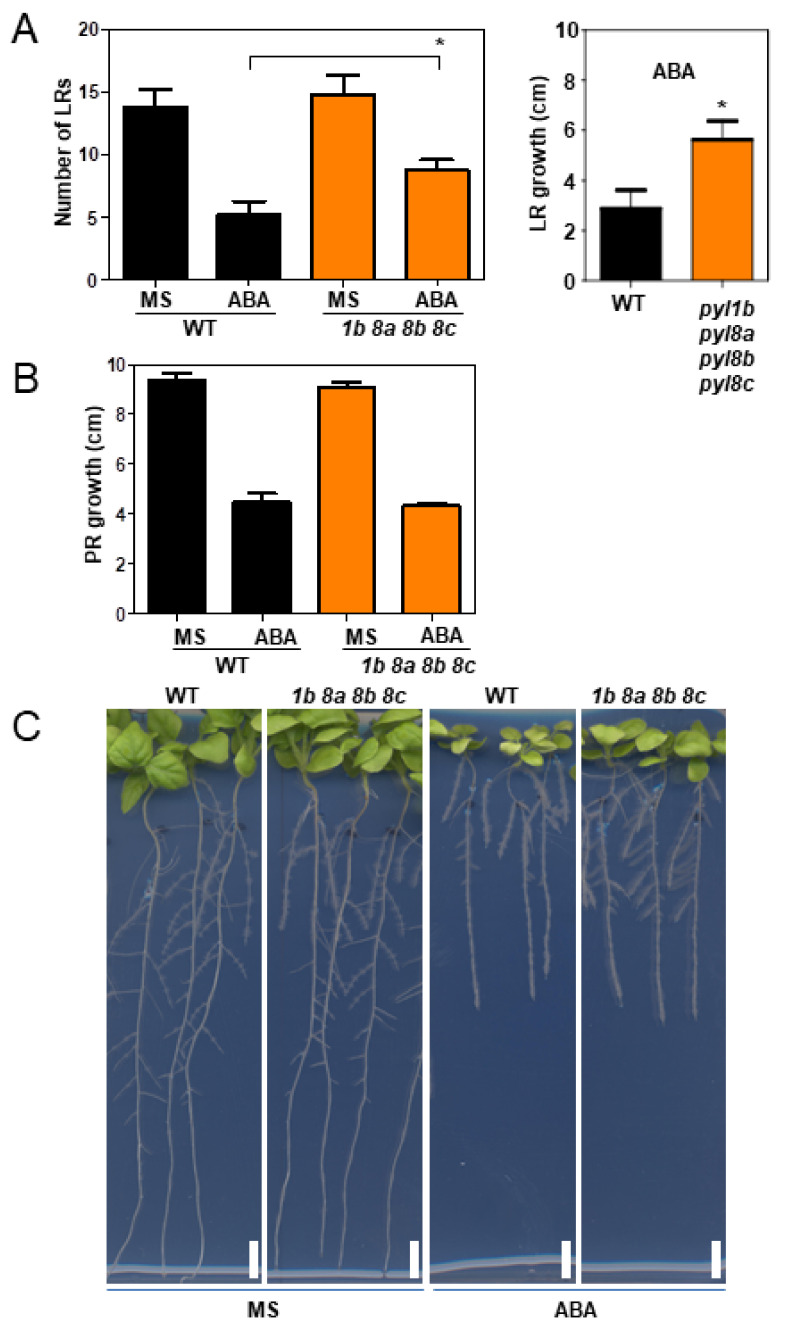
ABA-mediated inhibition of LR formation is reduced in *pyl1b pyl8a pyl8b pyl8c* relative to WT. (**A**) Inhibition of LR development and growth was scored in seedlings that were grown for 10 days in MS medium lacking or supplemented with 10 μM ABA. The number of lateral roots as well their length was scored using Image J. (**B**) PR growth was similar in the *pyl1b pyl8a pyl8b pyl8c* and WT plants. (**C**) Representative photographs of seedlings that were grown for 7 days in MS medium and then transferred to medium lacking or supplemented with 10 μM ABA. The *pyl1b pyl8a pyl8b pyl8c* and WT plants show differential growth of LRs in ABA medium. The asterisk indicates *p* ≤ 0.05 (Student’s *t*-test) when comparing data of mutant lines with non-transformed WT plants in the same assay conditions.

**Figure 7 cells-11-00795-f007:**
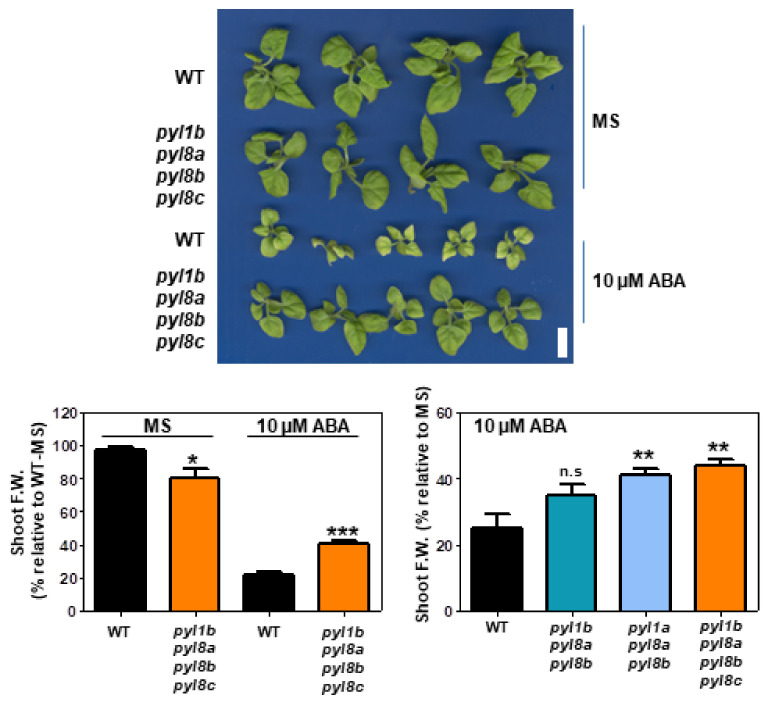
ABA-mediated inhibition of leaf growth is reduced in *pyl1b pyl8a pyl8b pyl8c* relative to WT. Leaf growth was scored in seedlings (n = 10) that were transferred from MS medium to medium lacking or supplemented with 10 μM ABA and grown for an additional 16 days. Total fresh weight (FW) of the mutant was measured and expressed as a percentage of FW relative to the WT in MS medium (100%, left panel) or in MS medium supplemented with ABA relative to the growth in MS medium lacking ABA (right panel). The asterisk indicates: * *p* ≤ 0.05, ** *p* ≤ 0.01 and *** *p* ≤ 0.001 (Student’s *t*-test or ANOVA followed by Dunnet’s test) when comparing data of mutant lines with non-transformed WT plants in the same assay conditions.

**Table 1 cells-11-00795-t001:** Description of the mutant alleles and location of the mutations in the coding sequence (CDS).

Allele	Mutation	Location in CDS
*pyl1a-1*	(+1) C	165/678
*pyl1a-2*	(Δ24) CACTGAGTTCCACACCTACCGAGT	150/678
*pyl1b-1*	(+1) C	165/678
*pyl1b-2*	(Δ1) C	165/678
*pyl8a-1*	(+1) T	46/528
*pyl8a-2*	(Δ2) GA	46/528
*pyl8b-1*	(+1) T	46/528
*pyl8c-1*	(+1) G	69/558
*pyl8c-2*	(Δ4) GGAG	69/558

**Table 2 cells-11-00795-t002:** Description of the biallelic (BA) or homozygous (HOM) single mutants obtained and their combination as high-order mutants.

	*PYL1a*		*PYL1b*		*PYL8a*		*PYL8b*	*PYL8c*	
High-Order Mutant	*pyl1a-1*	*pyl1a-2*	*pyl1b-1*	*pyl1b-2*	*pyl8a-1*	*pyl8a-2*	*pyl8b-1*	*pyl8c-1*	*pyl8c-2*
** *pyl1b pyl8a pyl8b pyl8c* **			BA	BA	HOM		HOM	BA	BA
** *pyl1b pyl8a pyl8b* **			BA	BA	HOM		HOM		
** *pyl1a pyl8a pyl8b* **	HOM				HOM		HOM		
** *pyl1a pyl1b pyl8a pyl8b* **		HOM		HOM		HOM	HOM		

**Table 3 cells-11-00795-t003:** Measurement of stomatal conductance (Gs) and transpiration rate (E) in mock- and 50 μM ABA-treated WT and mutant *Nb* plants. Stomatal conductance (mol m^−2^ s^−1^), transpiration rate (mol m^−2^ s^−1^), and leaf temperature (T leaf; °C) were measured. Each value is the mean (n = 5) of measurements performed in the leaves (4th leaf from the apex) of different plants. NS: non-significant differences within each treatment. For each genotype, an asterisk (*) indicates significant differences between control and ABA-treated plants.

		gs		E		T Leaf	
Mock	WT	0.100		2.1		25.9	
	1a 8a 8b	0.110		2.3		25.7	
	1b 8a 8b	0.106		2.2		25.8	
	1b 8a 8b 8c	0.103		2.0		25.9	
		NS		NS		NS	
ABA	WT	0.057	*	1.3	*	26.2	*
	1a 8a 8b	0.054	*	1.2	*	26.3	*
	1b 8a 8b	0.053	*	1.2	*	26.3	*
	1b 8a 8b 8c	0.061	*	1.4	*	26.2	*
		NS		NS		NS	

## Data Availability

The RNA-seq data reported in this article are available in the NCBI database (Bioproject PRJNA770890).

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
