# Peer review of "PYL1- and PYL8-like ABA Receptors of Nicotiana benthamiana Play a Key Role in ABA Response in Seed and Vegetative Tissue"

_cells, 2022, doi:10.3390/cells11050795_

Round 1
Reviewer 1 Report
The MS present a lot of interesting data but there is a lack of clear explanations and inaccuracies.
In fact:
Nicotiana benthamiana, N. tabacum and Arabidopsis thaliana must be written in Italic
Figure 2. It is indicated the inclusion of a total of 13 AtPYLs but at line 69 the member of Arabidopsis PYL family seems to be 14, one is missing? Moreover the Figure is easier to understand by adding (in italics) the abbreviations At and Nb before PYLx and omitting the sequence codes.
Lines 151-153. Since the different gene families are mentioned, it would be useful to describe in detail the differences between the families at the protein level.
Figure 3. Fig.3b should be moved in Supplementary materials; in addition, it is difficult to understand such Figure in comparison to Figure S1A.
Figure S1A. what is the meaning of “pb”?
Lines 156-159. It is not clear why seven genes were chosen for editing, not being all the most highly expressed genes chosen, nor were the same gene number per family.
Table 1. Not all 7 edited genes are included: an explanation indicating that mutants for the subfamily II were not obtained is necessary for ease of reading.
Figure 4A. Since the five mutated genes (proteins) belong to two different subfamilies, a comparison with Arabidopsis genes (proteins) belonging to the same two families is appropriate; thus, one element is missing between AtPYL7-AtPYL10.
In conclusions, it should be pointed out that, although interesting, the results are preliminary given the complexity of the Nb genome; that is, was it worth expecting and attempting editing for 23 genes?
Author Response
We appreciate and acknowledge the comments of the referees, and provide a point-per-point response. According to comments/suggestions of referee 1 (see response below), we provide modified versions of:
Figure 2, Figure 3b, Figure 4a, and Supplemental Figure 1a.
A modified version of the manuscript, including tracking changes, has been attached

Reviewer 2 Report
This manuscript describes functional analysis of the abscisic acid (ABA) receptors in Nicotiana benthamiana, which is one of the tobacco species. The ABA receptors in tobacco were previously identified by genomic analysis, but no functional analysis had been performed. In addition, although ABA receptors have been isolated and functionally analyzed in various plants, the reports on knockout mutant lines of the ABA receptors are limited to model plants such as Arabidopsis and rice. This manuscript has reported the traits of mutant lines of ABA receptor genes, which are mainly expressed in Nicotiana benthamiana. The quadruple mutant of NbPYL1a, 8a, 8b and 8c receptors showed the reduced sensitivity to ABA in the shoot and roots and was alleviated ABA-induced growth inhibition. In other words, NbPYL1 and MbPYL8 are involved in ABA-dependent growth inhibition. On the other hand, the transpiration rate of the quadruple mutant line was not altered, suggesting the involvement of other ABA receptors in the guard cells. This work will provide important insight on the phytohormone ABA research. In summary, in my opinion, this study is of sufficient quality to be accepted by Cells.
Minor comment
- Why did author exclude the Arabidopsis AtPYL6 gene in the phylogenetic tree analysis? I could not understand this.
Author Response
We appreciate and acknowledge the comments of the referee
Minor comment
- Why did author exclude the Arabidopsis AtPYL6 gene in the phylogenetic tree analysis? I could not understand this.
Thanks. We have corrected this point in the new Figure 2
Round 2
Reviewer 1 Report
The only remaining suggestion is tu use colors for histograms giving an identical specific color to each sample.
Author Response
We appreciate and acknowledge you very much for the comment. The manuscript has markedly improved thanks to this and previous comments.
We have changed the color of the histograms in Figures 5, 6, and 7, as well as FigS3. Now we have used and maintained a distinctive color for each genetic background in the different figures. The legend of each figure indicates the corresponding treatment for each genetic background.
This manuscript is a resubmission of an earlier submission. The following is a list of the peer review reports and author responses from that submission.